# Primary Chronic Sclerosing Osteomyelitis: A New Diagnostic Tool

**DOI:** 10.3390/diagnostics13233571

**Published:** 2023-11-29

**Authors:** Anne-Sophie Lacagne, Laurence May, Marie Nicod Lalonde, John O. Prior, Martin Broome

**Affiliations:** 1Department of Oral and Maxillofacial Surgery, Lausanne University Hospital, 1005 Lausanne, Switzerlandmartin.broome@chuv.ch (M.B.); 2Department of Nuclear Medicine and Molecular Imaging, Lausanne University Hospital, 1005 Lausanne, Switzerland; marie.nicod-lalonde@chuv.ch (M.N.L.); john.prior@chuv.ch (J.O.P.)

**Keywords:** primary chronic sclerosing osteomyelitis, three-phase bone scintigraphy, anti-granulocyte antibody scintigraphy

## Abstract

Aims: Primary chronic sclerosing osteomyelitis is a rare and complex pathology and remains a diagnostic and therapeutic challenge. Our aim is to show our experience with a new diagnostic tool. Material and Methods: Four patients aged from 26 to 67 were referred to the department of oral and maxillofacial surgery of University Hospital CHUV in Lausanne between January 2010 and December 2018 for chronic mandibular pain without infectious signs nor symptoms. All patients underwent three-phase bone scintigraphy and anti-granulocyte antibody scintigraphy. Results: Three-phase bone scintigraphy demonstrated radiotracer uptake at the zone of pain, whereas anti-granulocyte antibody scintigraphy showed no uptake, thus rendering an infectious origin unlikely. Conclusion: A combination of the two different scintigraphies should be considered in order to guide the clinician in the diagnosis of primary chronic sclerosing osteomyelitis, thus preventing patients from undergoing unnecessary imagery and useless treatment, and also allowing an early diagnosis.

Primary chronic sclerosing osteomyelitis (PCSO) is a rare disease characterized by chronic osseous nonbacterial inflammation with no known etiology. It was initially described by Carl Garré in 1893 [1] as an osteomyelitis of the sclerosing type, with osseous deformation and enlargement but without suppuration, sequester or fistula. Patients describe principally chronic pain that is difficult to control, unilateral swelling, and sometimes trismus without symptoms or signs of infection for more than 4 weeks [2]. If the disease not only affects bone but also skin and joints, it is considered to be SAPHO syndrome (synovitis, acne, pustulosis, hyperostosis and osteitis) [3].

Radiologic characteristics are nonspecific. Early stages may show periosteal apposition associated with osteolytic lacunar images, and later stages can reveal a dense sclerotic bone with or without bone deformation. Histologically, there are no specific findings for primary chronic sclerosing osteomyelitis. Generally, medullary fibrosis, chronic inflammation and subperiosteal reaction can be found [4]. Blood markers are equally nonspecific. Bacterial cultures are often negative and are performed to exclude infectious osteomyelitis [5,6].

Treatment of PSO is divided into three steps, beginning with anti-inflammatory drugs. The second step is per os steroid administration, and the third step involves intra-venous bisphosphonates [7]. Intravenous bisphosphonate treatment protocol in our department consists in pamidronate administration of 15 mg for 3 days. If improvement in the symptoms, but not their disappearance, is noted, two more days of treatment are added. In case of recurrence of pain and other symptoms, the same procedure is repeated after 3 months [8]. Physiotherapy combined with medical treatment is highly recommended (Figure 1, Figure 2, Figure 3, Figure 4 and Figure 5).

Surgical treatment, such as decortication or even more invasive segmental mandibulectomy, has been described. However, it shows a high recurrence rate, even after additional procedures [9,10].

Diagnosis of this disease remains challenging. PCSO remains a diagnosis of exclusion using a combination of clinical and radiological arguments. An early and simple procedure for the diagnosis of PCSO is the combination of three-phase bone scintigraphy and anti-granulocyte antibody scintigraphy, as shown in these clinical cases. Three-phase bone scintigraphy is a diagnostic imaging tool that can be used for the diagnosis of osteomyelitis [11].

Bone scintigraphy uses diphosphonate molecules labelled with ^99m^Tc, which accumulate in the remodelling bone via incorporation into calcium hydroxyapatite. Bone radiotracer uptake depends on local blood flow and the extent of bone remodeling (mostly on the osteoblastic activity). Bone diseases such as osteomyelitis show increases in vascularization (hyperemia) and increases in radiotracer uptake due to the increase in bone turnover. A valuable advantage of this procedure is that these metabolic modifications are seen often weeks before classic radiological images show any lytic or sclerotic lesions.

Scintigraphy performed with anti-granulocytes antibodies labelled with Tc-99 m has an accuracy close to 90% for the detection of osteomyelitis [11].

The antibody binds specifically to granulocytes in vivo without altering their function. The marked granulocytes then migrate to the infected tissue due to chemiotactic attraction. Accumulation of these granulocytes in the infected tissue is dependent on the site of infection, the type and virulence of the pathologic agents, previous antibiotic or steroid treatment, and the quality of vascularization.

Overall, anti-granulocytes antibody scintigraphy has a good sensitivity (84%) and specificity (80%) for the detection of chronic infection in the peripheral skeleton.

In conclusion, the combination of the two different scintigraphies should be considered to help the clinician support the diagnosis of PCSO, thus preventing the patients from undergoing unnecessary imaging and useless treatments, while also allowing early diagnosis and adequate therapy.

## Figures and Tables

**Figure 1 diagnostics-13-03571-f001:**
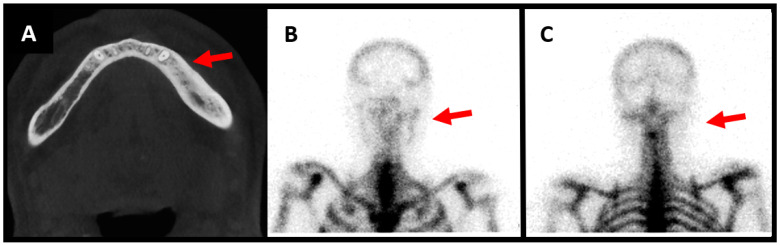
Male patient aged 57 years old, presenting with left mandibular pain associated with swelling ongoing for 1 year. He underwent extraction of tooth 35 prior to the pain. He notices improvement of the pain with nonsteroidal anti-inflammatory drugs, but recurrence of pain and swelling as soon as he stops medication. No infectious symptomatology is described by the patient and on clinical evaluation we notice no infectious signs and no hypoesthesia of the mandibular nerve. Surgical biopsy shows bone neoformation and fibrous reorganization. There is no inflammatory syndrome in laboratory results. CT scan demonstrates osteocondensation with aspects of sclerotic bone ((**A**), red arrow). The three-phase osseous scintigraphy shows alteration in the three phases, with hyperfixation of the tracer on the left mandibular horizontal bone compatible with a diagnosis of osteomyelitis ((**B**), red arrow). The scintigraphy with anti-granulocytes antibodies shows no hyperfixation of the antibodies in either early or late images, thus excluding an infectious osteomyelitis ((**C**), red arrow). The symptoms resolved after long-term treatment with nonsteroidal anti-inflammatory drugs.

**Figure 2 diagnostics-13-03571-f002:**
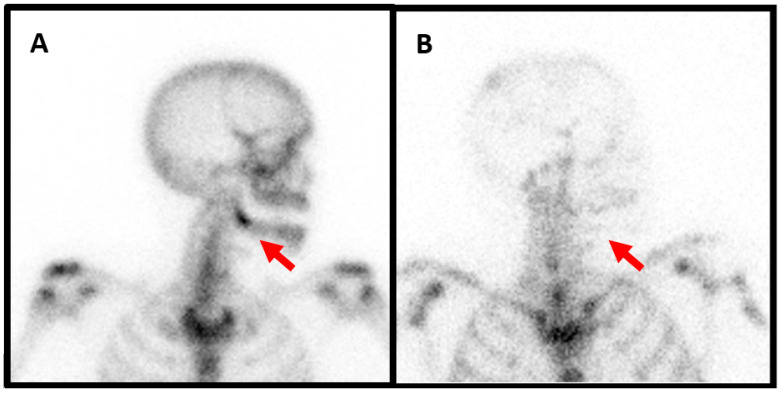
Female patient, 47 years old, presenting with recurrent right mandibular pain and with no improvement despite various analgesic treatment attempts. She developed avulsion of the tooth 48 one year ago. Biological results and bone biopsy show no infectious signs. Three-phase bone scintigraphy demonstrates hyperfixation of the tracer at the zone of the tooth extraction on right mandibular angle compatible with beginning of osteomyelitis ((**A**), red arrow). Anti-granulocytes antibodies scintigraphy shows no hyperfixation, thus excluding an infectious origin ((**B**), red arrow). This patient too needed treatment with intravenous of bisphophonates after insufficient response to long-term nonsteroidal anti-inflammatory drugs and per os steroid administration. After 2 years, the pain is controlled with occasional relapses, but without the need for surgical intervention.

**Figure 3 diagnostics-13-03571-f003:**
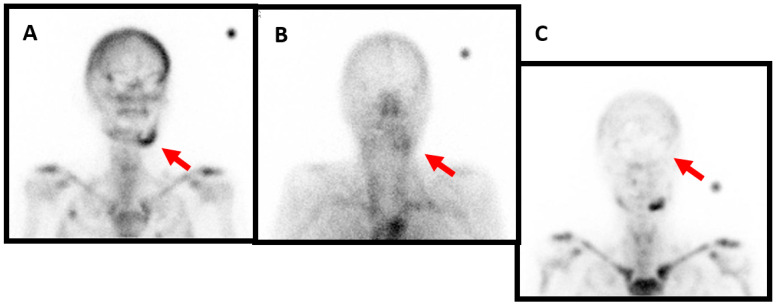
Female patient, 73 years old, who has been diagnosed with mandibular osteochemonecroses. She has been under bisphosphonate treatment for over 3 years for metastatic breast cancer and presents with mandibular pain, swelling and intraoral bone exposure. The profile image acquired 4 h after the intravenous injection of anti-granulocyte antibodies shows marked radiotracer uptake in the left mandibula ((**A**), red arrow). The blood pool phase (anterior view) of the bone scan shows left mandibular hyperemia ((**B**), red arrow). Likewise, the delayed image of the bone scan again shows marked radiotracer uptake ((**C**), red arrow). For this patient, the treatment plan involves antibiotic treatment, sequestrectomy, or possible mandibular resection and reconstruction with a fibula free flap depending on the residual bone quality.

**Figure 4 diagnostics-13-03571-f004:**
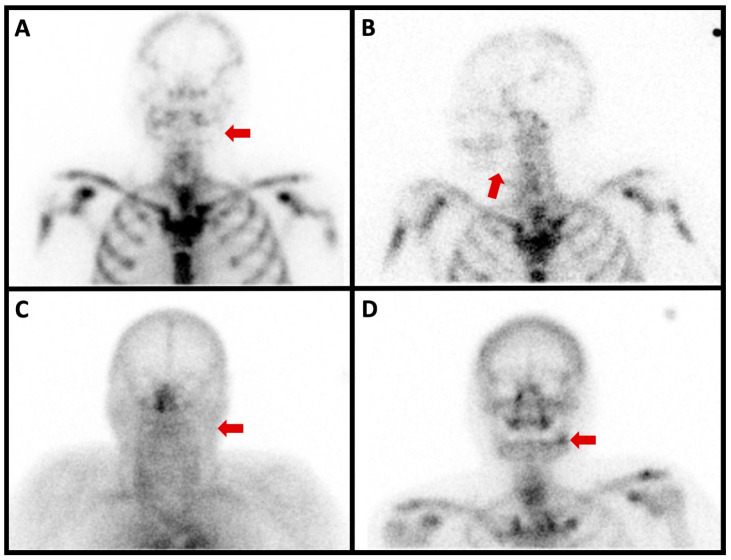
Planar scintigraphy imaging of a 67-year-old patient who had the extraction of tooth 38 with transient improvement. A few months later, the same pain recurred and was difficult to control with multiple analgesic attempts. Biological results showed no infectious signs. The anterior planar image acquired 6 h after the intravenous injection of anti-granulocyte antibodies showed no radiotracer uptake in the left mandibula ((**A**), red arrow). Likewise, no radiotracer uptake was seen in the left mandibula on the anterior left profile image acquired 24 h after the radiopharmaceutical injection ((**B**), red arrow). The blood pool phase (anterior view) of the bone scan showed no left mandibular hyperemia ((**C**), red arrow). On the other hand, the delayed image of the bone scan (anterior view) showed marked radiotracer uptake ((**D**), red arrow).

**Figure 5 diagnostics-13-03571-f005:**
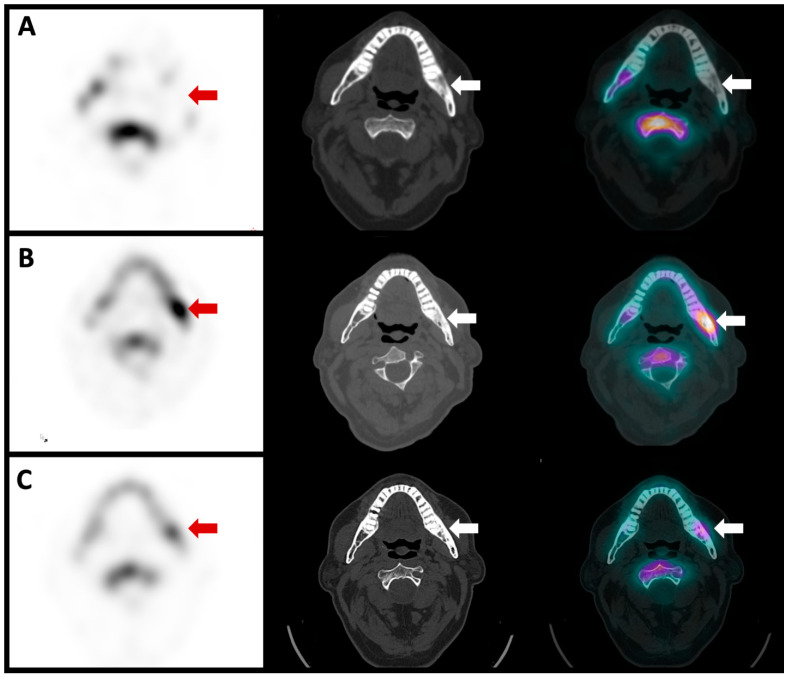
Single-photon emission computed tomography (SPECT), computed tomography (CT) and fusion images of the same patient showed no anti-granulocyte antibody uptake in the left mandibula ((**A**), red and white arrows). The images of the CT scan showed reactive osteosclerosis with no lytic or soft tissue collection suspicious of active infection (middle image, red and white arrow). The bone scan SPECT/CT showed intense radiotracer uptake in the left mandibula ((**B**), red and white arrows), indicating active osteoblastic activity. The patient was treated us biphosphonates and a follow-up bone scan SPECT/CT was repeated two years later, which showed a marked decrease in left mandibular radiotracer uptake ((**C**), red and white arrows).

## Data Availability

Data is contained within the article.

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
