# Peer review of "Primary Chronic Sclerosing Osteomyelitis: A New Diagnostic Tool"

_diagnostics, 2023, doi:10.3390/diagnostics13233571_

Round 1

Reviewer 1 Report

Comments and Suggestions for Authors

Chronic nonbacterial osteomyelitis (chronic sclerosing osteomyelitis ) is a steril inflammatory osteitis,that most commonly develops in the long bones,mandibular lesions are found in 1,5-3 % of disease foci in patients and it is poorly characterised in the maxillofacial surgery lierature.It is a really a diagnostic and  therapeutic challenge.The new diagnostic tool ,presented in this manuscript is important and interesting.

Author Response

Good evening

thank you very much for your interest in our manuscript and time for reviewing.

Reviewer 2 Report

Comments and Suggestions for Authors

The article presents interesting descriptions of the diagnosis and treatment of four cases involving patients suffering from chronic sclerosing osteomyelitis. Managing osteomyelitis in the jaw presents a significant challenge for maxillofacial surgeons, given the multitude of therapeutic and diagnostic approaches available, which lack a standardized treatment protocol.

The article is well-structured and informative, but a few minor revisions are recommended. To enhance clarity, it is advisable to replace "avulsion" with "extraction" in the terminology used. Additionally, the description of bisphosphonate treatment should include specific details such as the drug used, the dosage administered, and the treatment duration to improve the local condition.

Furthermore, the inclusion of recent publications would be beneficial. Incorporating articles like: "Is Operative Management Effective for Non-Bacterial Diffuse Sclerosing Osteomyelitis of the Mandible?" and "Application of pamidronate disodium for the treatment of diffuse sclerosing osteomyelitis of the mandible: A clinical study" would enrich the article with up-to-date insights and perspectives.

Comments on the Quality of English Language

Minor editing of English language required

Author Response

Good evening,

thank you for your interesting reviewing.

1. The terminology "avulsion" has been replaced by extraction.

2. Our iv bisphosphonate treatment protocol has been added on page 5, line 82-86.

3. I also added the two more recent references you mentionned as well as two further references: 

-Yildirim T D, Sari I. SAPHO syndrome : current clinical, diagnostic and treatment approaches.

-Van de Meent M M, Pichardo S E C, Appelman-Dijkstra N M, Van Merkesteyn J P R. Outcome of different treatments for chronic diffuse sclerosing osteomyelitis of the mandible: a systematic review of published papers.

Thank you very much for your time, best regards

Laurence May